# STARS-FL: Accelerating Federated Learning over Heterogeneous Mobile Devices via Spatial-Temporal Aware Reconfigured Subnetworks

## Abstract

Federated learning (FL) on mobile devices faces challenges from inherent computing and communication heterogeneity across devices. Subnetwork FL training offers a promising solution by assigning each device the largest feasible subnetwork extracted from the global model for local training. However, existing subnetwork training methods rely on static subnetwork assignments across time and uniform extraction ratios across layers. They overlook (i) the dynamic requirements of local contributions across FL training in temporal domain and (ii) the varying importance of different layers within subnetworks in spatial domain, both of which strongly affect the FL training performance. In this paper, we propose to accelerate FL training over heterogeneous mobile devices via spatial and temporal aware reconfigured subnetworks (STARS-FL). Different from existing approaches, STARS-FL leverages Fisher Information to identify critical learning periods and enables mobile devices to adjust their subnetworks across the FL training process correspondingly. Further, from the spatial domain, STARS-FL introduces a novel layer-wise subnetwork width adjustment mechanism, which enables each device to reconfigure layer widths adaptively based on its layer-specific computational and communication overheads, its real-time computing/communication conditions and potential straggler effects. Compared with state-of-the-art subnetwork methods, our experiments demonstrate that STARS-FL effectively speed up FL training while maintaining competitive learning accuracy.

## 1 Introduction

Driven by rapid advances in mobile GPU hardware, federated Learning (FL) McMahan et al. (2017) has shifted its focus from conventional data center environments to mobile devices Li et al. (2021a); Lim et al. (2020). This shift, coupled with FL's privacy-preserving nature, has enabled applications such as keyboard predictions Hard et al. (2018), health event monitoring Xu et al. (2021), and so on. In order to harness the immense potential of Federated Learning (FL) on mobile devices, researchers must tackle the considerable challenges due to the inherent heterogeneity of real-world mobile devices, which differ in computing power, network conditions, and local data distributions Lai et al. (2021). Many existing FL studies assume a model-homogeneous setting, where global and local models have identical model architectures across all clients. However, as devices are constrained to train models within their own capabilities, developers must choose between excluding lower-tier devices, which introduces training bias Bickel et al. (1975), or maintaining a low-complexity global model to support all clients, reducing accuracy Cho et al. (2021); Ye et al. (2020). Such an issue is intensified with the growing popularity of large models Liu et al. (2023); Kuang et al. (2024), which makes training on mobile devices even more challenging. In addition, mobile devices have relatively mediocre performances due to the constrained computing resources, along with their slow and unstable wireless connections. When compared to GPU clusters with stable, high-speed Internet connections, such inferior factors of mobile devices may lead to substantial latency in FL training Chen et al. (2022) and can significantly harm the performance of related applications.

To overcome the challenges of model-homogeneous federated learning (FL), recent research has focused on training models of varying sizes across diverse mobile devices and exploring methods for aggregating these heterogeneous models during FL training. Subnetwork training, as demonstrated by approaches such as width-based subnetwork generation in Federated Dropout Wen et al. (2021) and HeteroFL Diao et al. (2021), and depth-based generation in DepthFL Kim et al. (2023), has proven effective in enabling mobile devices to train smaller subnetworks derived from a large global model. These methods also provide solutions for aggregating subnetworks across diverse devices. By customizing subnetwork architectures to match each device's capability, subnetwork training improves adaptability to mobile devices with varying computational and communication resources. However, existing static subnetwork approaches cannot capture the dynamic requirements of local contributions across FL training in temporal domain and the varying importances of different layers within subnetworks in spatial domain, both of which may affect the FL training performance, especially the training latency and learning accuracy.

From the temporal domain, fast global model convergence has dynamic requirements for local training contributions during FL training. Recent studies have highlighted the existence of critical learning periods (CLPs) Achille et al. (2018); Yan et al. (2022) during FL training, where the final test accuracy is highly related to the performance in these CLPs. During CLPs, it will be good for mobile devices to employ larger-sized subnetworks to provide more local training contributions. Apart from CLPs, the required local contributions from FL clients may be less, so that smaller-sized subnetworks can be adopted by mobile devices to reduce both computing and communication delays. Thus, it is essential to characterize CLPs and dynamically reconfigure the subnetwork sizes for mobile devices aware of the global model's needs across time.

From the spatial domain, current subnetwork efforts typically use uniform extraction ratios across all layers of the model, ignoring the distinct impacts of different layers on FL performance. As we know, different layers in DNN models capture features at varying levels of the sample's underlying patterns while exhibiting different degrees of parameter redundancy. Uniform subnetwork extraction may allocate resources to parameter-heavy yet uncritical layers, increasing computational and communication costs with very limited FL performance gains. That necessitates spatial-aware subnetwork reconfiguration, i.e., layer-wise subnetwork width adjustment, for FL over heterogeneous mobile devices.

To address those challenges above, in this paper, we propose STARS-FL, a spatial and temporal aware subnetwork reconfiguration approach to accelerate FL training over heterogeneous mobile devices. STARS-FL enhances conventional subnetwork extraction strategies (based solely on device capabilities) through holistic consideration of both training dynamics and system efficiency. By dynamically reconfiguring subnetwork sizes according to CLPs in the temporal domain and adjusting layer-wise subnetwork widths in the spatial domain, STARS-FL achieves accelerated convergence while maintaining model accuracy. Our major contributions are summarized as follows.

- We define a global CLP evaluation criterion for FL based on Fisher Information to guide the dynamic scaling of subnetwork sizes in the temporal domain.

- We propose a layer-wise width adjustment policy that accounts for layer-specific computational and communication overheads, allowing mobile devices to locally reconfigure the trainable parameters of each layer in the spatial domain.

- We unify temporal and spatial subnetwork adjustments into a single reconfiguration function that incorporates mobile devices' real-time transmission and computational rates, which helps mitigate straggler-induced delays in FL convergence.

- We develop a STARS-FL prototype and evaluate its performance with extensive experiments. The experimental results validate that our design can remarkably reduce the latency for FL training over heterogeneous mobile devices without sacrificing learning accuracy.

## 2 PRELIMINARIES

### 2.1 SUBNETWORK-BASED FL

Given a wireless network consisting of $I$ mobile devices that collaboratively train a deep neural network (DNN) using FL, the local model parameters are denoted as $W_1, \cdots, W_i, \cdots, W_I$. Sub-

Table 1: Comparison among different subnetwork configuration methods. (ResNet18@CIFAR10)

| Method | Full Model | Fixed-ratio Subnetwork | Spatial-aware Subnetwork |
|---|---|---|---|
| Parameters | 42.6M | 10.7M | 3.4M |
| FLOPS | 330.2M | 83.3M | 78.1M |
| Accuracy | 88.4 | 83.2 | 82.8 |

network training Diao et al. (2021) is an approach to deal with system heterogeneity in FL training, which allows each device to extract subnetworks of varying sizes from the global model and perform local training. Let $\mathcal{W}^P = W^1, W^2, \ldots, W^p, \ldots, W^P$ represent a set of candidate subnetworks available for selection by mobile devices, where $P$ denotes the number of complexity levels. A higher level $p$ corresponds to a smaller subnetwork, satisfying $W^P \subset W^{P-1} \subset \cdots \subset W^1$. Subnetworks are extracted from the global model by reducing the width of hidden channels using predefined shrinkage ratios, which are selected to align with the respective computing capabilities of the mobile devices. Let $s \in (0, 1]$ denote the hidden channel shrinkage ratio; then the relationship between subnetwork sizes is given by $|W^p|/|W_g| = s^{2(p-1)}$. In each FL training round, the server aggregates the updates from these heterogeneous subnetworks using the following rule:

$$W_g = W_g^P \cup (W_g^{P-1} \setminus W_g^P) \cup \cdots \cup (W_g^1 \setminus W_g^2), \tag{1}$$

where $W_g^p$ is the global subnetwork of the level $p$. For each subnetwork level, the parameters are averaged across the devices assigned to that particular subnetwork size. This aggregation ensures that each parameter is averaged using contributions from devices whose assigned subnetwork includes that parameter, which enables global aggregation and FL training across subnetworks of varying sizes.

## 2.2 Critical Learning Periods in FL

Critical learning periods in FL refer to specific phases during training when the neural network undergoes significant changes in how it learns and organizes information. Notably, the information in the weights does not increase monotonically during training; instead, it experiences a rapid growth phase followed by a reduction. Accordingly, the adjustment of subnetwork sizes in FL is reasonable to align with such training dynamics, which can be characterized using fisher information as revealed in recent findings Achille et al. (2018). Fisher information essentially provides a second-order approximation of the Hessian of the loss function Amari & Nagaoka (2000); Martens (2014), which offers insights into the curvature of the loss landscape around the current weights. Specifically, we adopt the Federated Fisher Information Matrix (FedFIM) from Yan et al. (2022). In round $r$, FedFIM can be formally calculated by:

$$FI_i^r = \mathbb{E}_{x_i \sim \mathcal{X}_i} \mathbb{E}_{\hat{y} \sim p_W(\hat{y}_i|x_i)} [\nabla(x_i, \hat{y}_i) \nabla(x_i, \hat{y}_i)^\mathsf{T}]. \tag{2}$$

Here, $x_i$ represents the input data, and $y_i$ denotes the corresponding output label for device $i$. The model parameters are denoted by $W$, while $p_W(\hat{y}_i|x_i)$ represents the approximate posterior distribution. The empirical distribution of the $i$-th device's local data is represented by $\mathcal{X}_i$. The gradient of the loss function for a data point $(x, y)$ is denoted as $\nabla(x, y) = \frac{\partial}{\partial W} \ell(x, y; W)$. Notably, $\hat{y}_i$ is treated as a random variable rather than a true label, with its distribution governed by $p_W(\hat{y}_i|x_i)$. To simplify computation, we approximate the full FedFIM by using its trace, which can be efficiently calculated as:

$$\mathrm{tr}(FI_i^r) = \mathbb{E}_{x_i \sim \mathcal{X}_i} \mathbb{E}_{\hat{y} \sim p_W(\hat{y}_i|x_i)} [|| \nabla(x_i, \hat{y}_i)||^2]. \tag{3}$$

## 3 Motivation

**Ignoring spatial awareness in subnetwork extraction.** Existing subnetwork training methods, such as HeteroFL Diao et al. (2021), typically extract subnetworks by applying a uniform reduction ratio across all layers. However, this approach overlooks the spatial characteristics of prevalent deep

learning models, which have varying parameter redundancy across layers and the distinct computational/communication costs introduced by different layers. As a result, applying a uniform reduction ratio can overemphasize the importance of some parameter-heavy layers while underestimating their associated costs, leading to suboptimal resource utilization.

To illustrate this point, we conducted a preliminary study comparing full-model FL, fixed-ratio subnetwork FL, and spatial-aware subnetwork FL. Specifically, the fixed-ratio one extracts a subnetwork with a reduction ratio of 0.5 across all layers, while the spatial-aware one applies a lower ratio for deeper layers. Table 1 summarizes the results, which show that our spatial-aware subnetwork structure achieves comparable test accuracy to the fixed-ratio one while significantly reducing parameters and FLOPs. These findings demonstrate the necessity of incorporating spatial awareness into subnetwork extraction, which allows for layer-wise adjustments that align with both parameter redundancy and resource constraints for a more time-efficient FL process.

**Ignoring temporal awareness in subnetwork extraction.** Existing subnetwork training methods employ a fixed extraction ratio throughout the FL training process, overlooking the existence of CLPs and their dynamic requirements on the subnetwork sizes. The presence of Critical Learning Periods (CLPs) and their evolving requirements for subnetwork sizes. We find that information in the weights experiences a phase of rapid growth followed by a reduction, even as test accuracy continues to improve. This behavior has been consistently observed across various tasks and network architectures Achille et al. (2018); Yan et al. (2022). These findings underscore the significance of the training contributions during CLPs, as insufficient learning during this period can irreversibly impact the final accuracy of the FL process, regardless of additional training efforts.

At the start of the FL training process, contributions from all devices are valuable, making smaller subnetworks preferable for faster computation and transmission of local updates. During the CLP, where the network undergoes substantial learning and reorganization, the server requires more precise local model updates to drive convergence. Finally, as training nears convergence, the server has already accumulated most of the critical contributions from the devices. Accordingly, large or full-sized subnetworks are no longer necessary, as they may introduce significant latency while offering limited benefits for further training. Thus, dynamically adjusting subnetwork sizes based on temporal training phases is essential to balance efficiency and performance in FL.

## 4 STARS-FL DESIGN

### 4.1 STARS-FL OVERVIEW

STARS-FL features dynamic subnetwork reconfiguration to accelerate FL training over heterogeneous mobile devices. As reflected in the subsection titles, STARS consists of (i) global temporal-aware (TA) subnetwork reconfiguration, (ii) local spatial-aware (SA) subnetwork reconfiguration, (iii) integrated spatial-temporal aware (STA) subnetwork reconfiguration, and (iv) straggler-resilient STA subnetwork updates.

STARS-FL generally follows the client-server architecture of traditional FedAvg protocol but mainly differs in the subnetwork reconfiguration. As shown in Fig. 1, the outlined workflow is as follows: ① The server initializes and assigns a small-sized subnetwork to each device, tailored to its wireless transmission rate and computing capacity, to ensure a target per-round duration. ② Mobile devices locally train their assigned subnetworks and upload the updated gradients to the server. ③ The server aggregates the received updates. ④ The server exploits global Fisher information to identify CLPs and decide temporal-aware subnetwork size reconfiguration for the next FL round. ⑤ Based on the updated global model and temporal-aware subnetwork guidance, each device further applies spatial-aware layer width adjustments to reconfigure its subnetwork and conducts the local training. Steps ②–⑤ are repeated until FL global model converges.

### 4.2 GLOBAL TA SUBNETWORK RECONFIGURATION

We propose to reconfigure the subnetwork sizes for mobile devices aligned with the evolving training requirements of the global model across time. Specifically, the training dynamics of the FL global model can be characterized using global Fisher information, which is expressed as follows.

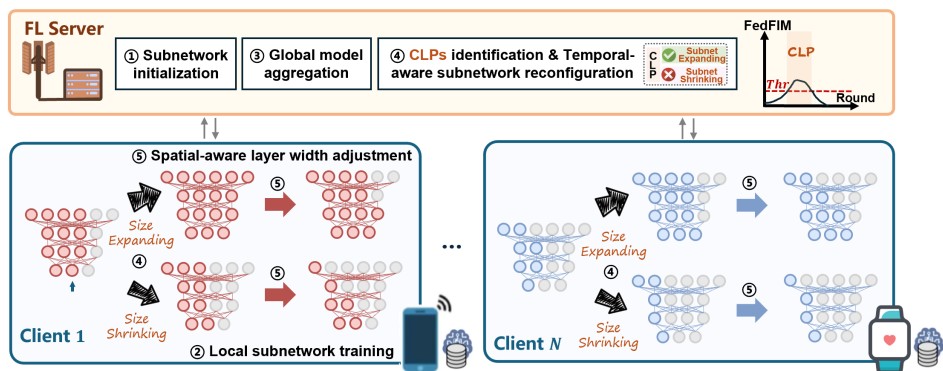

Figure 1: The sketch of the STARS-FL procedure.

$$TD_r = \frac{1}{N} \sum_{i}^{N} \sum_{d=0}^{D-1} |\mathbf{B}_i| \sqrt{\frac{1}{|\mathbf{B}_i|} \sum_{k \in \mathbf{B}_i} FI_i^{r-d}(k)^2}, \tag{4}$$

where $N$ is the total number of mobile devices and $\mathbf{B}_i$ represents the sample batch for the $i$-th device. Here, we average the FIM-based local training efficiency across all devices during training to estimate the global model training status, which provides a temporal understanding of the global model convergence's requirements. To measure local training dynamics for each individual device, we adopt a window-averaged expression with a window size of $D$. This helps to smooth out short-term fluctuations in the training process, providing a more stable view of local fisher information's changes. After receiving the updated gradients and FIMs from all mobile devices, the server averages the aggregated gradients as FedAvg and calculates the $TD_r$ using Eq. (4).

By comparing $TD_r$ with a predefined threshold $thr$ as shown in Eq. (7), the server determines whether to increase or decrease the local subnetwork sizes and sends the global TA subnetwork reconfiguration strategies together with the updated global model to the devices for the next-round training.

Regarding computational complexity, our STARS-FL introduces overhead primarily from Fisher Information computation. As formalized in Eq. 3, we approximate Fisher Information via the squared norm of gradients, which can be parallelized with SGD operations. As a result, our design yields negligible computational overhead, which is also verified by empirical results in Table 2 showing no significant runtime increase versus standard FL.

### 4.3 LOCAL SA SUBNETWORK RECONFIGURATION

Upon receiving the guidance from the server to reconfigure subnetwork size (i.e., expand or shrink), each device determines a specific layer-wise width adjustment strategy considering the spatial structure of the model. We denote $\Delta W_{i,l}^{r,+}$ as the expansion adjustment in the $l$-th layer width compared to the previous training round, $W_{i,l}^{r-1}$. For the scenario of width expanding, the layer-wise width expansion adjustment is defined as follows:

$$\Delta W_{i,l}^{r,+} = \lfloor \min\{\frac{\alpha_l^{-1}}{\sum_k \alpha_k^{-1}}, \Delta\} \rfloor, \tag{5}$$

where $\Delta \in (0, 1]$ is a developer-based adjustment interval that constrains the maximum step size for width expansion per layer.

$\alpha_l = N_l / \sum_l N_l$ represents the proportion of the $l$-th layer's parameters $N_l$ relative to the total number of parameters in the entire model (i.e., $\sum_l N_l$). A larger $\alpha_l$ indicates that the $l$-th layer is more parameter-dense, which in turn results in bigger computational and transmission overheads for local training and gradient updates. To prevent disproportionately high overhead, the expansion pace for such layers is moderated, as reflected in Eq. (5).

Similarly, for the width shrinking scenario, the adjustment is defined as follows:

$$\Delta W_{i,l}^{r,-} = \lfloor -\min\{\alpha_l, \Delta\}\rfloor. \tag{6}$$

In this case, parameter-dense layers will shrink more rapidly due to their typically higher parameter redundancy.

### 4.4 STA Subnetwork Reconfiguration

STA subnetwork reconfiguration integrates global TA subnetwork guidance and local SA subnetwork adjustment in the previous two subsections. Such a STA subnetwork reconfiguration is jointly done by the server, which is aware of the global model's temporal training dynamics, and mobile devices, which are aware of the local subnetwork's spatial importance/redundancy of layer-wise parameters. The STA subnetwork reconfiguration policy is formulated as follows.

$$\Delta W_{i,l}^r = \begin{cases} \lfloor \min\{\frac{\alpha_l^{-1}}{\sum_k \alpha_k^{-1}}, \Delta\}\rfloor, & \textbf{if } TD_r \geq thr; \\ \lfloor -\min\{\alpha_l, \Delta\}\rfloor, & \textbf{if } TD_r < thr. \end{cases} \tag{7}$$

In essence, when the $TD_r$ is high, indicating it is a CLP, the participating devices opt to increase the size of their subnetworks. For those parameter-heavy layers, the width increment is moderated to preserve training efficiency. Conversely, when $TD_r$ is low, signifying a not that critical period, the mobile devices reduce their subnetwork sizes. For parameter-heavy layers, STA subnetwork reconfiguration reduces more width to further alleviate computational and communication burdens.

### 4.5 Straggler-Resilient STA Subnetwork Updates

Besides temporal-spatial awareness, our STARS-FL design makes subnetwork updates straggler-resilient. In particular, for the $i$-th device in round $r$, the number of weights in the $l$-th layer of its subnetwork is updated based on spatial- and temporal-aware width adjustment $\Delta W_{i,l}^r$ as follows.

$$W_{i,l}^r = W_{i,l}^{r-1} + \left(\frac{\bar{t}}{t_i}\right)^{\gamma \cdot \mathrm{sgn}(\Delta W_{i,l}^r)} \Delta W_{i,l}^r. \tag{8}$$

Here, $\mathrm{sgn}(\Delta W_{i,l}^r)$ represents the sign of $\Delta W_{i,l}^r$, which indicates whether the layer width should expand or shrink. $t_i$ is the per-round duration of device $i$, and $\bar{t}$ denotes the average duration among all the devices in the previous training round, which serves as the bottleneck reference. Thus, $\frac{\bar{t}}{t_i}$ tells whether mobile device $i$ is prone to becoming a straggler. A larger $\frac{\bar{t}}{t_i}$ indicates that device $i$ is significantly faster, which allows it to retain more parameters in its subnetwork and contribute more to FL training. The hyperparameter $\gamma$ controls the sensitivity of subnetwork width adjustments to the device's relative speed. A larger $\gamma$ puts more emphasis on mitigating the impact of stragglers, while a smaller $\gamma$ places less emphasis on the straggler issue. When the subnetwork size increases (i.e., $\mathrm{sgn}(\Delta W_{i,l}^r) > 0$), we aim to limit the extent of width expansion for slower mobile devices to prevent excessive computational and transmission burden. Conversely, when the subnetwork size decreases (i.e., $\mathrm{sgn}(\Delta W_{i,l}^r) < 0$), we prioritize reducing the width slightly more for faster mobile devices to maintain a balanced contribution across devices. Such an adaptive subnetwork updating mechanism ensures that it effectively accommodates device heterogeneity.

Let $N_l$ denote the total number of parameters on $l$-th layer. After obtaining the number of trainable parameters $W_{i,l}^r$ for layer $l$, we can further calculate the subnetwork extraction ratio of the layer via:

$$\tilde{W}_{i,l}^r = \frac{W_{i,l}^r}{N_l}, \tag{9}$$

where $\tilde{W}_{i,l}^r \in (0, 1]$ represents the proportion of parameters trained in layer $l$. To facilitate implementation on mobile devices, these values are quantized into discrete candidate layer widths:

$$\hat{W}_{i,l}^r = \begin{cases} W^1, & \text{if } \tilde{W}_{i,l}^r \geq s; \\ W^2, & \text{if } \tilde{W}_{i,l}^r \in [s^2, s); \\ \cdots, & \cdots \\ W^p, & \text{if } \tilde{W}_{i,l}^r \in [s^p, s^{p-1}); \\ \cdots, & \cdots \\ W^P, & \text{if } \tilde{W}_{i,l}^r < s^{P-1}, \end{cases} \tag{10}$$

where $s$ is the shrinkage ratio of layer widths and $|W^p|/|N_l| = s^{2(p-1)}, \forall l, W^p \in \mathcal{W}^P$. In every FL training round, each device can perform subnetwork configuration by applying the above width adjustments across all layers. This design ensures subnetwork sizes dynamically adapt to the changing requirements of local training contributions, while the layer-wise width adjustments are tailored to both the devices' capabilities and the model's structural characteristics. This helps improve both training efficiency and resource utilization in FL training.

The convergence of STARS-FL can be established by extending the theoretical framework in Su et al. (2025) through a layer-wise neuron region redefinition of subnetworks. This analysis demonstrates that STARS-FL converges under adaptive subnetwork size scheduling.

## 5 EXPERIMENTAL SETUP

### 5.1 IMPLEMENTATION TESTBED

Our implementation comprises an FL server equipped with an NVIDIA A40 and 20 heterogeneous mobile devices, including MacBook Pro 2018, NVIDIA Jetson Xavier, NVIDIA Jetson TX2, NVIDIA Jetson Nano, and Raspberry Pi 4, with four devices of each type in the system. These devices have significantly different CPU frequencies and memory, thereby effectively reflecting the computational heterogeneity in real-world scenarios. Communication between the server and devices occurs over LTE, Bluetooth 3.0, and Wi-Fi 5 networks. For subnetwork configuration, we set the hidden channel shrinkage ratio to $s = \frac{1}{2}$ and use 5 subnetwork size levels. The width shrinkage ratios for the five levels are $1$, $\frac{1}{4}$, $\frac{1}{16}$, $\frac{1}{64}$, and $\frac{1}{256}$, respectively. To simulate data heterogeneity, we use a balanced non-IID data partition Li et al. (2021b), where $\sigma$ represents the level of non-IIDness, corresponding to the number of classes on each device. For our experiments, we use the default setting of $\sigma = 2$. Additionally, we set the layer width adjustment interval to $\Delta = 0.125$ and the threshold for identifying critical learning periods (CLPs) to $thr = 1 \times 10^{-5}$, unless otherwise specified.

### 5.2 TASKS AND BASELINES

We conduct our experiments with two common FL tasks. For the image classification task, we train a CNN model (2 convolutional layers and 1 fully connected layer) on the MNIST dataset and a ResNet18 model on the CIFAR10 dataset. For the human activity recognition (HAR) task, we train a CNN model on the HAR dataset Gupta et al. (2022). We compare our STARS-FL approach with the following baselines: (1) **FedAvg** McMahan et al. (2017), where all the devices train with full-sized models. (2) **HeteroFL** Diao et al. (2021), which employs fixed subnetwork extraction ratios throughout FL training, uniform across model layers, with subnetwork sizes matching devices' full computational and communication capacities. (3) **FedDropout** Wen et al. (2021), which generates subnetworks by choosing the neurons at random. (4) **FedRolex** Alam et al. (2024), which extracts subnetworks in a rolling way across FL training rounds. (5) **TARS-FL**, a variant of STARS-FL incorporating only temporal-aware subnetwork reconfiguration. (6) **SARS-FL**, another variant of STARS-FL incorporating only spatial-aware subnetwork reconfiguration.

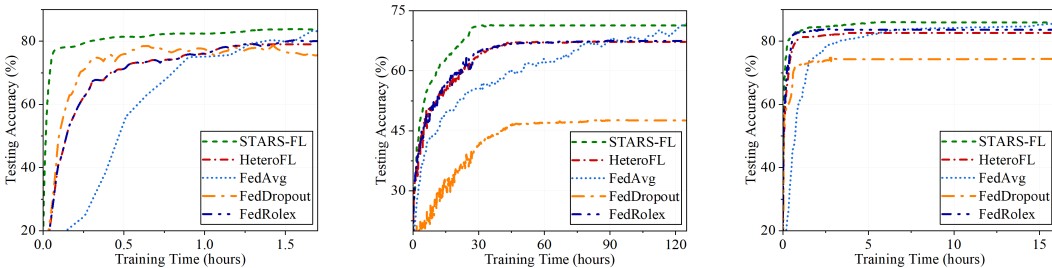

Figure 2: Performance comparison of different FL training approaches under various learning tasks. Figures from left to right are CNN@MNIST, ResNet18@CIFAR10, and CNN@HAR with non-IID ($\sigma = 2$) datasets.

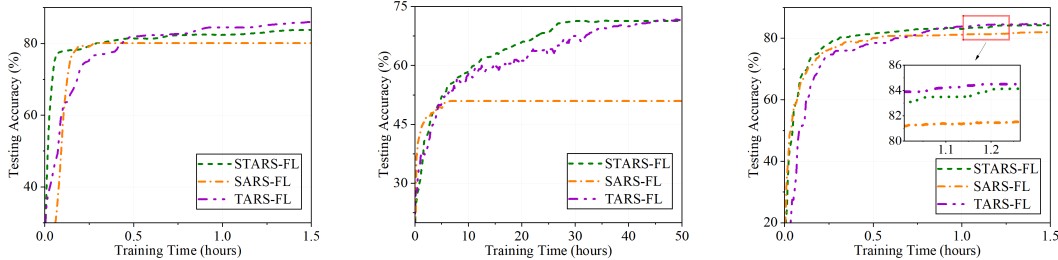

Figure 3: Performance analysis under STARS-FL, TARS-FL, and SARS-FL. Figures from left to right are CNN@MNIST, ResNet18@CIFAR10, and CNN@HAR with non-IID ($\sigma = 2$) datasets.

# 6 EVALUATION AND ANALYSIS

## 6.1 LATENCY EFFICIENCY AND LEARNING PERFORMANCE

We begin by evaluating the latency efficiency of STARS-FL across different FL tasks. As shown in Fig. 2, our STARS-FL consistently achieves significant training speedup across these tasks without sacrificing learning accuracy. Compared with FedAvg, STARS-FL accelerates the FL training to the target testing accuracy by approximately 4.23x, 3.21x, and 3.16x for CNN@MNIST, ResNet18@CIFAR10, and CNN@HAR tasks, respectively. By incorporating temporal and spatial-aware subnetwork reconfiguration, STARS-FL assigns larger subnetworks to devices during CLPs and reduces the widths of parameter-dense layers to minimize communication and computational overheads. This dynamic adaptation ensures that local subnetworks are well-suited to varying communication and computational conditions, as well as the evolving requirements of FL training at different stages. In contrast, HeteroFL, FedRolex, and FedDropout employ a static fixed-size subnetwork assignment policy, which may increase transmission time without yielding significant accuracy improvements. As a result, STARS-FL achieves faster convergence and higher final accuracy compared to the fixed-size methods, demonstrating its superior FL training efficiency and overall performance.

## 6.2 ABLATION STUDY

We compare STARS-FL with variants that feature either temporal-aware or spatial-aware subnetwork reconfiguration. The experiments are conducted on CNN@MNIST, ResNet18@CIFAR10, and CNN@HAR FL tasks, with results shown in Fig. 3. In the presence of CLPs during early training, SARS-FL's lack of temporal awareness causes devices to train with smaller subnetworks compared to STARS-FL. While SARS-FL benefits from reduced transmission time due to spatial-aware design, it cannot adjust subnetwork sizes according to the different training phases, resulting in lower accuracy and slower convergence than STARS-FL. Similarly, TARS-FL, which lacks spatial-aware layer width adjustment and may assign unnecessarily large widths to parameter-heavy layers, experiences longer transmission delays. Although TARS-FL occasionally achieves slightly higher accuracy than STARS-FL, its convergence speed is slower. These findings confirm that STARS-FL effectively balances convergence speed and accuracy.

## 6.3 SENSITIVITY ANALYSIS

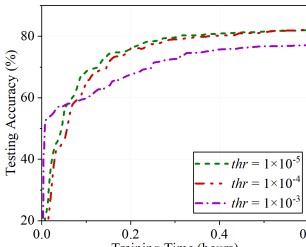 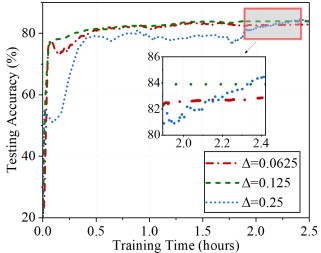 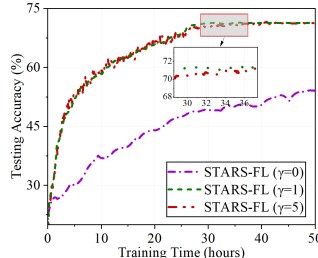

Figure 4: Sensitivity analysis under $thr$, $\Delta$, and $\gamma$ values with non-IID ($\sigma = 2$) datasets.

We conduct sensitivity analysis of STARS-FL's performance under different $thr$, $\Delta$, and $\gamma$ values, and present the results in Fig. 4.

The hyperparameter $thr$ is a threshold used to identify CLPs. As shown in Fig. 4(a), selecting a larger threshold ($thr = 1 \times 10^{-3}$) makes the devices less likely to switch to larger subnetworks during CLPs, which leads to faster early training with reduced computational and transmission latency, but resulting in subpar final accuracy due to smaller subnetworks that limit learning capacity. In contrast, with smaller thresholds ($thr = 1 \times 10^{-4}$ and $thr = 1 \times 10^{-5}$), devices are more likely to select larger subnetworks during CLPs, which, although causing slower convergence in the early stages, leads to higher final accuracy. Therefore, appropriately setting $thr$ can improve time-to-accuracy performance by balancing early training speed with the need for sufficient learning capacity during CLPs.

The hyperparameter $\Delta$ governs the maximum step size for subnetwork width expansion or shrinkage. As shown in Fig. 4(b), larger intervals like $\Delta = 0.3$ can induce abrupt changes in subnetwork size reconfigurations across devices, potentially destabilizing local training and impeding FL convergence. Conversely, overly conservative intervals (e.g., $\Delta = 0.0625$) may lead to insufficient adaptation of subnetwork dimensions to evolving training requirements, ultimately degrading model performance. Our experimental results identify $\Delta = 0.125$ as achieving superior performance through balanced adjustment granularity under our settings, enabling faster convergence while maintaining model accuracy. Practical implementations may benefit from dynamically tuning $\Delta$ through real-time monitoring of accuracy progression, allowing step size adaptation to specific task demands and system states.

The hyperparameter $\gamma$ controls how responsive subnetwork width adjustments are to a device's relative training speed compared with others. Our experimental results in Fig. 4(c) reveal that setting $\gamma = 0$ essentially ignores slower devices' limitations. This configuration allows slower devices to frequently process oversized subnetworks, creating significant training delays. Simultaneously, faster devices become constrained from expanding their subnetwork sizes efficiently during CLP. Consequently, devices may fail to achieve optimal subnetwork sizes, leading to prolonged training times and reduced final accuracy. Notice that using $\gamma = 5$ achieves very similar performance to $\gamma = 1$, while it marginally overcompensates for slower devices' impacts and results in slightly inferior model outcomes.

## 7 CONCLUSION

In this paper, we have proposed STARS-FL, a novel subnetwork-based FL design that integrates temporal and spatial awareness into subnetwork reconfiguration. In the temporal domain, STARS-FL leverages Fisher Information to identify critical learning periods and dynamically adjusts subnetwork sizes to align with evolving training dynamics. In the spatial domain, it introduces a layer-wise subnetwork width adjustment mechanism, enabling adaptive configuration of layer widths based on layer-specific computational and communication constraints. Through extensive experiments, we have demonstrated the superior performance of STARS-FL in achieving training speedup while maintaining competitive accuracy.

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

## A    APPENDIX

The appendix contains supplementary experimental results.

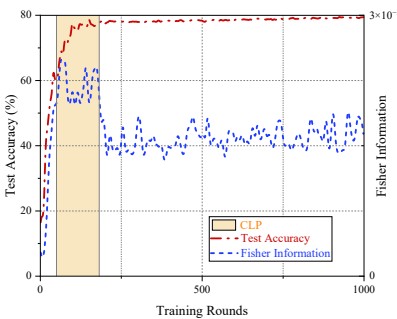

Figure 5: The relation between Fisher Information and test accuracy.

Table 2: Performance comparison under different data heterogeneity levels $\sigma$ (Task: CNN@MNIST).

| $\sigma$ | 2 | 5 | 10 | 2 | 5 | 10 |
|---|---|---|---|---|---|---|
| Metric | Final Accuracy | | | Latency (hours) | | |
| FedAvg | 89.1 | 98.9 | 99.0 | 13.36 | 15.42 | 12.34 |
| HeteroFL | 79.0 | 98.4 | 98.8 | 1.50 | 2.50 | 1.62 |
| TARS-FL | 85.9 | 98.9 | 99.0 | 1.78 | 2.35 | 1.88 |
| SARS-FL | 80.1 | 98.2 | 98.8 | 0.53 | 1.01 | 0.95 |
| STARS-FL | 83.6 | 98.7 | 99.0 | 1.35 | 1.48 | 1.29 |

In Fig. 5, we show the relationship between CLPs and test accuracy, with the information embedded in weights quantified using FedFIM from Eq. (3).

We further evaluate the impact of data heterogeneity on STARS-FL's performance using CNN@MNIST as an example. We examine cases where each device has $\sigma = 2, 5$ or $10$ classes, where the data distribution is IID if $\sigma = 10$, i.e., every device has all classes. The results are shown in Table. 2. As the non-IID level increases, FL training typically achieves lower final accuracy. However, STARS-FL maintains accuracy close to that of FedAvg (within a 6% loss), while achieving faster convergence by 9.8x, 10.4x, and 9.5x for $\sigma = 2, 5$ or $10$, respectively. In extreme non-IID scenarios ($\sigma = 2$), HeteroFL's accuracy drops from 89.1 to 79, while STARS-FL achieves an accuracy of 83.6 with a notable 10% reduction in convergence time. This improvement is attributed to STARS-FL's ability to capture training dynamics and model structural characteristics for sub-network reconfiguration, which enables a remarkable enhancement in FL training efficiency under various data heterogeneity scenarios.

