# OpenReview forum: "STARS-FL: Accelerating Federated Learning over Heterogeneous Mobile Devices via Spatial-Temporal Aware Reconfigured Subnetworks"
_ICLR.cc/2026/Conference — ICLR 2026 Conference Withdrawn Submission_

### Official Review · Reviewer_72UD · 2025-10-28

**Soundness:** 2
**Presentation:** 2
**Contribution:** 2
**Rating:** 2
**Confidence:** 4

**Summary:**

This paper proposes a subnetwork-based federated learning approach designed to accelerate FL training through adaptive subnetwork configuration. The method integrates temporal and spatial awareness into subnetwork reconfiguration: it utilizes Fisher Information to identify critical learning periods and dynamically adjust subnetwork size over time, and further introduces a layer-wise width adjustment mechanism that enables adaptive layer configuration.

**Strengths:**

1.	**Practical focus:** The idea of subnetwork-based FL is practically relevant for improving training efficiency in resource-constrained environments.
2.	**Real-device evaluation:** The paper evaluates the proposed method using real devices, which adds credibility and realism to the experimental validation.

**Weaknesses:**

1.	**Limited novelty:** The concept of subnetwork-based FL is not new, and there exist many prior works in this area. The claimed motivation—that existing subnetwork training methods rely on static subnetwork assignments over time and uniform extraction ratios across layers—is not entirely accurate, as several existing methods already employ dynamic subnetwork.
2.	**Incomplete baseline coverage:** The experimental comparison lacks several relevant subnetwork-based FL baselines that should have been included for a fair evaluation. Moreover, these works (e.g., [1-7]) are not discussed in the related work section, making it difficult to contextualize the proposed contributions.
3.	**Simplistic evaluation setup:** The datasets used are relatively simple, and the performance improvement over previous methods is modest. As a result, the empirical evidence is not strong enough to demonstrate a clear advantage of the proposed approach.

**Questions:**

1.	Compared with the following previous subnetwork-based FL methods, where does the novelty of the proposed approach lie? Could the authors also include performance comparisons with these methods in the evaluation?
2.	Could the authors provide evaluation results on more challenging and real-world datasets to better demonstrate the effectiveness of the proposed method?





[1] Wu, Feijie, et al. "FIARSE: Model-heterogeneous federated learning via importance-aware submodel extraction." Advances in Neural Information Processing Systems 37 (2024): 115615-115651.

[2] Wang, Haozhao, et al. "Feddse: Distribution-aware sub-model extraction for federated learning over resource-constrained devices." Proceedings of the ACM Web Conference 2024. 2024.

[3] Li, Ang, et al. "Hermes: an efficient federated learning framework for heterogeneous mobile clients." Proceedings of the 27th annual international conference on mobile computing and networking. 2021.

[4] Li, Ang, et al. "LotteryFL: Empower edge intelligence with personalized and communication-efficient federated learning." 2021 IEEE/ACM Symposium on Edge Computing (SEC). IEEE, 2021.

[5] Li, Ang, et al. "Fedmask: Joint computation and communication-efficient personalized federated learning via heterogeneous masking." Proceedings of the 19th ACM conference on embedded networked sensor systems. 2021.

[6] Horvath, Samuel, et al. "Fjord: Fair and accurate federated learning under heterogeneous targets with ordered dropout." Advances in Neural Information Processing Systems 34 (2021): 12876-12889.

[7] Luo, Junyu, et al. "Fedskel: efficient federated learning on heterogeneous systems with skeleton gradients update." Proceedings of the 30th ACM international conference on information & knowledge management. 2021.

---

### Official Review · Reviewer_JKoV · 2025-10-29

**Soundness:** 2
**Presentation:** 2
**Contribution:** 2
**Rating:** 2
**Confidence:** 5

**Summary:**

The paper introduces STARS-FL, a federated learning system for heterogeneous mobile devices that leverages spatial and temporal aware subnetwork reconfiguration. Unlike traditional subnetwork FL methods that use static, homogeneous subnetworks, STARS-FL dynamically adjusts subnetwork sizes in response to training phase (via Fisher Information-based detection of critical learning periods) and per-layer characteristics (spatial domain). The approach is evaluated with prototypes across multiple tasks and achieves substantial training acceleration.

**Strengths:**

1）The paper targets a well-known and practical problem. The core idea, that subnetwork assignments should be dynamic in both time (temporal) and model structure (spatial), is highly intuitive. Moreover, the preliminary study in Table 1, which shows a spatial-aware subnetwork achieving similar accuracy with fewer parameters than a fixed-ratio one, is a simple and effective motivator

2）A significant strength of the evaluation is its use of a physical, heterogeneous testbed. The experiments are not just simulations; they are run on a diverse set of real devices (MacBook, Jetson Nano/TX2/Xavier, Raspberry Pi)  over real networks. This makes the latency and speedup claims (e.g., in Figure 2 and Table 2) far more credible than simulation-only results.

**Weaknesses:**

1. The discussion of related work is incomplete, omitting several recent and highly relevant studies. The paper fails to situate itself against foundational work on depth-scaling mechanisms (e.g., FEDEPTH [1]), model-heterogeneous FL via distillation (e.g., FedMD [2]), and successive layer training [3]. This omission makes it difficult to assess the paper's true methodological advancement over the current state-of-the-art and undermines its novelty claims.
2. The paper provides no convergence proof for STARS-FL. It dismisses this critical requirement with a "hand-wavy" assertion that convergence "can be established by extending the theoretical framework in Su et al. (2025)" . This is insufficient. A complex method that dynamically reconfigures model architectures in a non-uniform, layer-specific way, and also changes this policy over time, requires a rigorous, non-trivial analysis to prove that the process converges at all.
3. The paper's premise of dynamically expanding subnetworks seems to misunderstand a primary motivation for methods like HeteroFL. Many heterogeneous FL approaches are designed to accommodate hard memory constraints, where a device has a fixed maximum feasible model size. The proposed method, which assumes devices can simply expand their subnetworks during a CLP, would fail in these common, memory-bound scenarios, as the device would crash when asked to load a subnetwork that exceeds its physical memory capacity
4. The evaluation lacks comparisons to several highly relevant subnetwork-based baselines (e.g., FEDEPTH [1]). The evaluation is restricted to small-scale datasets (especially MNIST) and relatively simple models (e.g., CNN, ResNet18) . The absence of trials on modern, large-scale FL tasks severely limits the paper's generalizability claims. Moreover, the experimental results (notably Figures 2–4) are primarily measured in terms of "Training Time" and "Testing Accuracy". The paper provides no detailed breakdown of per-device resource consumption (e.g., actual CPU/GPU utilization, memory footprint, or communication bytes).

[1] Memory-adaptive depth-wise heterogenous federated learning, AAAI, 2023

[2] Fedmd: Heterogenous federated learning via model distillation, Nips, 2019

[3] Aggregating capacity in fl through successive layer training for computationally-constrained devices, Nips, 2023

**Questions:**

See Weaknesses.

---

### Official Review · Reviewer_XxXs · 2025-10-31

**Soundness:** 2
**Presentation:** 2
**Contribution:** 2
**Rating:** 2
**Confidence:** 4

**Summary:**

This paper introduces STARS-FL, a novel federated learning (FL) framework designed to accelerate training across heterogeneous mobile devices by dynamically adjusting subnetwork sizes based on spatial and temporal awareness. It uses Fisher Information to identify critical learning periods (CLPs) for temporal adaptation and applies layer-wise width adjustments to account for spatial heterogeneity in model layers. STARS-FL also includes a straggler-resilient mechanism to handle slow devices.

**Strengths:**

- It jointly exploits temporal training dynamics (CLPs) and spatial layer heterogeneity, enabling more efficient subnetwork reconfiguration.

- Uses Fisher Information to accurately identify critical learning periods, allowing devices to dynamically scale subnetwork sizes for optimal learning contributions.

**Weaknesses:**

- The experiments focused on relatively small models (e.g., ResNet-18, shallow CNNs) . There is no assessment of how the proposed method scales to larger models (e.g., transformers).

-  The layer-wise importance metric is based solely on parameter counts, which may not accurately reflect actual computational or energy costs. This assumption could lead to suboptimal pruning decisions.

- While Fisher Information is used to guide subnetwork resizing, the required additional data (e.g., per-parameter Fisher values) may introduce significant communication overhead, particularly for large models. This cost is not thoroughly analyzed or compared with existing  techniques.

- The Non-IID evaluation is limited to mild data skew (σ = 2), and no results are shown for more challenging distributions.

**Questions:**

See the weaknesses.

---

### Official Review · Reviewer_32iK · 2025-10-31

**Soundness:** 2
**Presentation:** 2
**Contribution:** 2
**Rating:** 2
**Confidence:** 5

**Summary:**

This paper introduces a spatial-temporal aware subnetwork reconfiguration approach for federated learning over heterogeneous mobile devices, called STARS-FL. In contrast to existing subnetwork-based federated learning methods that employ static, uniform extraction ratios throughout training, this paper dynamically adjusts subnetwork sizes across both temporal and spatial dimensions. The key idea is to leverage Fisher Information to identify Critical Learning Periods (CLPs) during training and enable devices to expand their subnetworks during these critical phases, while simultaneously introducing layer-wise width adjustments that account for parameter density, computational costs, and redundancy across different layers. The framework also incorporates a straggler-resilient mechanism that adapts subnetwork adjustments based on each device's relative training speed. The authors evaluate their approach on real heterogeneous mobile devices across multiple tasks (CV and HAR) and compare with various federated learning baselines.

**Strengths:**

1. The paper identifies two key limitations of prior subnetwork-based FL methods: static subnetwork assignments that ignore temporal training dynamics, and uniform layer extraction ratios that overlook spatial characteristics of different layers.
2. The paper addresses federated learning over heterogeneous mobile devices, which is a practically significant scenario for real-world FL deployment.

**Weaknesses:**

# Weaknesses

1. **Limited adaptability to dynamic resource constraints**: Although the paper claims "subnetwork reconfiguration," the hidden channel shrinkage ratios are quantized into fixed discrete levels (s, s², s⁴, etc. as shown in Eq. 10), rather than being continuously adaptive. For memory-constrained devices, this fixed discretization may be either overly conservative (wasting resources) or overly optimistic (causing memory overflow), and cannot respond to real-time resource availability changes such as fluctuating memory or battery levels during training.

2. **Insufficient clarity in the local-global coordination mechanism**: The paper lacks detailed illustrations of how local spatial-aware adjustments and global temporal-aware guidance coordinate in practice. While Section 4.2 and 4.3 describe each component separately, the interaction between server-side CLP detection and client-side layer-wise adjustments remains unclear, making it difficult to understand the complete workflow.

3. **Limited novelty in subnetwork generation and unclear contribution breakdown**: Although the paper claims to introduce spatial and temporal awareness, the specific mechanisms (Fisher Information for temporal, parameter density ratios for spatial) are relatively straightforward applications of existing concepts. More critically, it is unclear how each dimension (spatial vs. temporal) individually contributes to the training acceleration; the ablation study shows combined effects but does not decompose the speedup contributions.

4. **Lack of analysis for partial client participation**: The paper does not address scenarios where clients do not participate in every round, which is common in practical FL. It is unclear whether the Fisher Information-based CLP detection and subnetwork size adjustments remain effective when only a subset of clients participate intermittently.

5. **Unclear scalability and stability guarantees under dynamic conditions**: The paper does not discuss how the proposed method handles large-scale FL with unstable training dynamics or how network configuration adapts when resource constraints vary significantly over time. The limitations of the spatial-temporal reconfiguration approach in highly dynamic environments are not characterized.

6. **Outdated experimental setup with limited model complexity**: The evaluation uses relatively small and outdated models (CNN on MNIST, ResNet18 on CIFAR10) which are insufficient in 2025. The paper lacks experiments on modern large language models or foundation model fine-tuning scenarios, which are increasingly relevant for federated learning applications.

7. **Insufficient hardware implementation details and questionable experimental credibility**: The paper provides minimal information about the actual hardware deployment. Claims such as "Communication between the server and devices occurs over LTE, Bluetooth 3.0 xxx" are confusing (how does Bluetooth 3.0 connect to a server?). Critical details are missing: resource distributions across heterogeneous devices, actual local training configurations, whether experiments are server-simulated or genuinely distributed, and concrete specifications of computational and memory constraints for each device type. Without this information, the experimental results cannot be verified or reproduced (even if they provide source code).

**Questions:**

see Weaknesses

---

### Note · Authors · 2025-11-18

I have read and agree with the venue's withdrawal policy on behalf of myself and my co-authors.